# Teleworking and Job Quality in Latin American Countries: A Comparison from an Impact Approach in 2021

Yanira Marcela Oviedo-Gil [1],* and Favio Ernesto Cala Vitery [2]

1   Econometría Consultores, Bogota 110231, Colombia
2   Faculty of Natural Sciences and Engineering, University Jorge Tadeo Lozano, Bogota 111711, Colombia; favio.cala@utadeo.edu.co
*   Correspondence: yanirovi@gmail.com

**Abstract:** This article studies the relationship between teleworking and job quality in 2021 for reference countries in Latin America, namely, Argentina, Brazil, and Colombia. Teleworking is an increasingly important alternative line of work; hence, it is relevant to investigate its influence on the quality of jobs. To this end the following points are addressed: (i) the definitions of teleworking and job quality are conceptually reviewed; (ii) the relationships between both concepts are identified in the literature; (iii) a comparable job quality index is constructed for each of the countries studied using factorial analysis, ensuring the objective nature of the data is considered rather than preconceived judgments; and (iv) the characteristics of job quality and the constructed index are compared in a controlled manner using a propensity score matching model. This research finds that teleworkers, compared to other workers with similar observable characteristics, have higher labor income in Colombia and Argentina. In addition, teleworkers work fewer hours per week and have similar levels of job stability across the three countries. Regarding teleworkers in Brazil and Colombia, a gap in social security coverage is observed and needs to be addressed.

**Keywords:** telework; job quality; work modalities





## 1. Introduction

Advances in information and communication technologies (ICT) have resulted in innovative ways of living. At present, it is possible to telework or work remotely supported by ICT. According to the International Labour Organization (ILO), in 2019, the existence of "home workers" was registered in most countries of the world. In 13 countries, home worker proportions of approximately 15% were registered (ILO 2020a).

Teleworking has been discussed since the 1950s and its theoretical advantages led Baruch to predict its widespread use (Baruch 2001; Jones 1957; Nilles et al. 1976; Toffler 1980; Kelly 1985; Baruch and Nicholson 1997). Nevertheless, extensive use has only recently occurred in the face of isolation measures due to the COVID-19 virus. After the beginning of the pandemic, the media and specialized entities, such as the Inter-American Development Bank (IADB), announced that teleworking experienced significant growth, along with the demand for specialized software to facilitate teleworking (IADB 2020).

By the end of 2021, the Organization for Economic Cooperation and Development (OECD) indicated that for every country with relevant data, an increase in teleworking was evident. For example, in Australia, France, and the United Kingdom, this increase was close to 47% in 2020, while in Japan it was 28%. In addition, the use of teleworking in highly digitized industries was reported to be, on average, 50% (OECD 2021). The analysis of teleworking trends, made by the OECD, suggests that teleworking's permanence depends on the balance made by workers and firms by sector, with respect to the convenience of maintaining it.

On this matter, the Brooking Institution envisions that teleworking will likely continue long after the pandemic. The argument is that the adoption of telecommuting has been

slower than many predicted due to rigid work cultures, as well as a lack of interest from employers in investing in the related technology and management practices necessary to operate a long-distance workforce. However, given the isolation measures associated with COVID-19, it was necessary to make such investments and create a more flexible business and work culture, thus resulting in a new valid production strategy (Guyot and Sawhill 2020). Therefore, teleworking is considered a labor innovation of recent times (Sánchez and Montenegro 2019; Castellano et al. 2017).

Due to all of the above, teleworking is a topic of public policy that has gained relevance and suggests that the world is facing a transformation that needs to be modeled, since it can positively or negatively influence relevant factors such as job quality, labor participation, economic growth, productivity, and quality of life, among others. To contribute to closing this research gap, this research studies what is meant by teleworking, what is meant by job quality, and how teleworking is related to job quality[1]. This facilitates the identification of lines of public policy to be recommended to take advantage of the potential advantages of teleworking and address its possible disadvantages.

Our findings are reported in this paper, which is structured as follows: first, an introduction is presented. Second, the definitions of teleworking and job quality are conceptually reviewed and the relationships between both concepts are identified in the literature. Third, the data and methodologies used are elaborated upon, which include the construction of indices through factorial analysis and the application of an impact evaluation technique. Fourth, data are preliminarily examined using general descriptive statistics and a job quality index is constructed for each of the countries studied using factorial analysis, ensuring the objective nature of the data is considered rather than preconceived judgments. In addition, the characteristics of job quality and the index are compared in a controlled manner using a propensity score matching model. Fifth, the results of the investigation are presented, which indicate that teleworkers, compared to other workers with similar observable characteristics, have higher labor income in Colombia and Argentina. In addition, teleworkers work fewer hours per week and have similar levels of job stability across the three countries. Regarding teleworkers in Brazil and Colombia, a gap in social security coverage is observed and needs to be addressed.

## 2. Teleworking and Its Relationship with Job Quality

### 2.1. Teleworking and Job Quality

To begin the analysis, we review the concepts of teleworking and job quality. Due to the recent interest in studying teleworking, the ILO has published some documents about the conceptual differences between "remote work," "telework," "work at home," and "home-based work" (ILO 2020a, 2020b). It is crucial to note that teleworking is a subcategory of "home work" which uses ICT to work remotely. A more specific concept extensively cited in the literature of multilateral agencies is "the use of ICT—such as smartphones, tablets, laptops and desktop computers—for the purposes of work outside the employer's premises" (Eurofound and ILO 2019, p. 5). There are also similar concepts adopted as national definitions for teleworking in Latin American countries (see Table 1).

Regarding the concept of job quality, there is no single accepted definition in the scientific literature, although there is agreement that job quality refers to every aspect of the job that is related to the well-being of the employees (Steffgen et al. 2020). In this sense, the OECD refers to job quality as the attributes of the job itself, independent of workers' characteristics (Cazes et al. 2015). A more comprehensive definition is established by the United Nations Economic Commission for Europe (UNECE). They state that job quality denotes the conditions of related employment not only as a source of income, but also as a source of social security, identity, and self-esteem (UNECE 2015).

In addition, Findlay et al. (2013) state that job quality is a multidimensional phenomenon wherein there is some consensus on its components, namely, earnings and benefits, the degree of job security, flexibility in working hours, the possibilities of using skills, degree of participation in decisions, and autonomy in tasks. Similarly, the ILOs

vision of "Decent Work" points out that the quality of a job should be reviewed through elements such as adequate income, stability and security at work, equal opportunities, work–family–personal life balance, and a safe work environment, among others (ILO 1999). Likewise, the G-20 plans to improve the quality of jobs in three main dimensions: quality of earnings, reducing labor market insecurity, and promoting good working conditions and a healthy society (G-20 2015).

**Table 1.** National definitions for teleworking in Latin America.

| Colombia | Argentina | Brazil |
|---|---|---|
| "Teleworking. It is a way of labor organization, which consists of the performance of paid activities or provision of services using information and communication technologies—ICT—for contact between the worker and the company, without requiring the physical presence of the worker in a specific job site"[2] (Law 1221 2008), article 2. | "( . . . ) a form of remote work, in which the worker carries out his activity without the need to physically present himself at the specific company or workplace. (...) [teleworking] It is carried out using information and communication technologies (ICT) and can be carried out at the worker's home or in other places or establishments outside the employer's home" Ministry of Labor, Employment and Social Security.[3] | "( . . . ) the provision of services predominantly outside the employer's premises, with the use of information and communication technologies that, by their nature, do not constitute external work"[4] (Da Silva 2020, p. 2). |

Source: (Law 1221 2008) and (Da Silva 2020).

Note that some of these characteristics of a job are objectively measurable, so they can be observed by a third party, while others are mediated by the perceptions of workers. Therefore, job quality can be analyzed from a subjective perspective when approached through the lens of job satisfaction, as psychologists do. In this regard, there is a wide range of recent studies in developed countries (Miglioretti et al. 2021; Azarbouyeh and Naini 2014; Sousa-Uva et al. 2021; Afonso et al. 2022; Antunes et al. 2022; Marx et al. 2021; Zöllner and Sulíková 2021). Nevertheless, Eurofound suggests measuring "( . . . ) observable job features that relate to meeting people's needs from work. The concept includes those characteristics of work and employment that have been proven to have a causal relationship with health and well-being" (Eurofound 2021, p. 4).

By using objectively measurable variables, the employment quality measurement strategy relies mainly on indices. These indices can be simple or compound, the latter being widely used in the literature given the multidimensional nature of the object of study (Costrell 1990), frequently based on the theoretical framework of the segmentation of labor markets. The use of job quality indices is reviewed in detail by Muñoz et al. (2011). Specifically, they built a job quality index for European countries. Their proposal is based on conceptualizing job quality as those aspects of the job that have a clear and direct impact on the well-being of workers, focusing on payment, intrinsic quality of work, health and safety, and work–life balance, assigning equal weight to each aspect and applying it to both independent workers and salaried workers.

In the literature there are several other job quality indices, especially in Europe. UN-ECE uses a 7-dimensional and a 12 sub-dimensional index[5] (UNECE 2015); the ILO index covers 10 substantive elements which are closely linked to the four strategic pillars of the Decent Work Agenda[6] (ILO 1999). The OECD index includes three key complementary dimensions of job quality: earnings, labor market security, and the working environment (Cazes et al. 2015). Steffgen et al. (2020) declare that there is still no general agreement regarding the dimensions or measurement of dimensions of job quality.

The richness of job quality dimensions may be limited by the availability of standardized data. This is more relevant in developing countries such as those of Latin America. Therefore, studies in the region focus on the components of income, social security coverage,

job stability, and workday. This is the case in a recent study considering the countries of Argentina, Bolivia, Brazil, Chile, Colombia, Ecuador, Mexico, Paraguay, Peru, and Uruguay. In the paper, the authors established a threshold for each dimension to determine whether a person is deprived or not within each dimension. They found that there are large differences in job quality among Latin American countries. While Chile presents the best results, Paraguay presents the worst, followed by Mexico, Bolivia, and Peru (Sehnbruch et al. 2020).

In Colombia, one of the first measurements of job quality was developed by Stefano Farné at the beginning of the 21st century. Labor income, affiliation with social security, working hours, type of contract, possibilities of promotion and training, workplace, and possibilities of unionization were included as dimensions of analysis (Farné 2003). Subsequently, updates and new proposals have been prepared for the measurement of job quality; among them, a study by Jiménez and Páez (2014) utilizing principal components methodology to find which dimensions can measure job quality correctly is highlighted. The significant dimensions they found were employment contract, adequate remuneration, affiliation with social security, and fair working hours.

In Argentina, job quality indices have been applied such as the Global Index of Working Conditions by the Center for Studies for Argentine Development (CENDA 2004) and the Labor Fragility Index (Kostzer et al. 2005). More recently, Born and Sacco (2016) built the Urban Labor Market Situation Index that summarizes labor situations of the economically active population with two large dimensions: precariousness and income from work. The index includes data from between 2003 and 2015. They discovered three periods of evolution of the job quality in Argentina. Between 2003 and 2007, they detected a marked increase due to lower unemployment rates and better incomes, between 2007 and 2011 they observed a period of moderate growth and, finally, witnessed stagnation between 2011 and 2015.

Multidimensional indices have also been used in Brazil to measure and study the quality of employment (Balsadi 2007; Do Nascimento et al. 2008). Recently, Paulo et al. (2021) did so with respect to rural employment and its dynamics, adding an analysis of a dynamic panel model to the results of the index. The authors conclude that rural employment is more precarious than urban employment, although there is evidence that the difference between them has been reducing over time. In addition, they found that since 2000, despite economic growth, the average quality of rural employment has not improved.

To study job quality, the predominant path taken by the scientific community has been in utilizing the theory of labor market segmentation (TLMS) because it helps us to understand the heterogeneous conditions of the working population, especially in developing countries. In this theory, there is a market segment characterized by highly productive jobs, those that are well-paid and with multiple benefits. This is often called the "formal sector." There is also another segment that generates jobs with low added value and poor working conditions called the "informal sector." The existence of this segmentation implies that there are barriers for workers when changing segment (Piore 1975; Bourguignon 1979; Anderson et al. 1987; Gittleman and Howell 1995; Maloney 1998).

The TLMS appears as an alternative to the Neoclassic Model, wherein perfect information, free mobility (no barriers to entry), and automatic adjustment are assumed. When there is segmentation, good and bad jobs coexist within the same economy. In the formal sector, there are barriers to entry due to various social, economic, and individual characteristics; meanwhile, in the informal sector, workers perform repetitive tasks and receive low income (Dickens and Lang 1991). According to Uribe et al. (2005), in this model, workers in the low-wage sector start out with the same human capital as those in the high-wage sector but end up with depreciated or obsolete human capital. That is, bad jobs tend to create low-quality workers. Therefore, because of low incomes, informal sector firms tend not to introduce labor-saving technologies and this leads to stagnant worker productivity.

The TLMS has been studied in Latin America, finding evidence of its presence in Colombia (Uribe et al. 2005, 2007; Urrea 2009; Uribe and Ortiz 2012), Argentina (Jiménez 2011; Waisgrais 2001; Arakaki 2017), and Brazil (Dalberto and Cirino 2018; Tannuri-Pianto

and Pianto 2016). Thus, the theory could be an appropriate theoretical framework for the analysis of job quality in the region.

It is also relevant to mention that, according to the ILO, teleworking is more difficult to implement when there is resistance from management, a lack of appropriate IT tools and devices, complicated operational processes, and lack of skills and support for teleworking teams. Therefore, having good practices in teleworking is challenging even for formal firms (ILO 2020c). Therefore, it would be reasonable to assume that the majority of teleworking, in a country with a segmented labor market, is associated with the formal sector of the labor market. This would imply that teleworking would present good relative working conditions.

*2.2. Potential Relationship between Teleworking and Job Quality*

In the academic literature, the quality of employment associated with teleworking has been studied mainly through the perception of employers and workers in multiple case studies, experiments, and opinion polls (Baines 1999; Benjumea-Arias et al. 2016; Las Heras and Barraza 2019). These kinds of studies have found benefits and challenges attributed to teleworking. Among them, the following stand out:

**Workday**. In an opinion poll in the United Kingdom, 38% of teleworkers during isolation reported that they worked more hours through this modality (McCulley 2020). In the European Union, 27% of teleworkers expressed their need to sacrifice personal time to meet the requirements of their jobs (Eurofound 2020), and some people reported that teleworking generated interference between work and home life (Eurofound and ILO 2019). Likewise, in the United States, a perception survey found that workers were busier, worked longer hours, and had more chaotic schedules with the arrival of the pandemic and when using telework (Reisenwitz 2020).

In contrast, performance measurement using goals or results is a priority in teleworking. This leads to more flexible working hours since the focus of the firms is not on the time worked, but on the productivity and results achieved by the staff (Caillier 2012; Madsen 2011). Studies in Europe have indicated that workers appreciate flexibility in the working day. They stated that teleworking allowed them to spend more time with family and reduced the costs associated with travel. Women were found to be more likely to assert that it was easier to reconcile work and personal life through teleworking compared to men (Eurofound and ILO 2019). An ILO analysis found that the flexibility of the working day is the main advantage of teleworking (ILO 2021; Eurofound and ILO 2019).

Along the same lines, in an analysis prior to the pandemic in the Latin American context, Delgado and Osio (2011) state that teleworking is associated with strengthening family ties and education at home, as well as a better quality of life due to less stress, less expenses in transportation and clothing, and better feeding conditions. Another study in the UK found that the line between work and home was blurred in telecommuting. However, full-time teleworkers expressed greater satisfaction with their jobs and working hours than occasional teleworkers or those who never teleworked (Wheatley 2012). Therefore, greater work–family balance could have a positive impact on the level of satisfaction and motivation, which leads to lower staff turnover, greater efficiency, and a reduction in the need for office space and the associated costs from the point of view of firms, as presented in Eurofound and ILO (2019).

Gentilin (2021) goes further and points out that the transition to the knowledge economy has favored the flexibility of other labor issues, such as monitoring, measurement and control, workplace, etc. In this sense, Bueno (2018) infers that millennial-generation people especially appreciate this type of flexibility, and although salary remains a priority, flexibility and culture in the workplace also have an impact, particularly regarding autonomy at work.

**Labor income**. In a study in Italy, it was found that jobs that can be carried out from home, not only teleworking, are more likely to earn low income, as well as to result in occupational accidents and occupational illness (Cetrulo et al. 2022). In another study

conducted in California (USA), the authors analyze the characteristics of teleworkers and their homes. They divide the employed into three categories, namely teleworker, home-based, and non-teleworker. They find that teleworkers have the highest household incomes among the three groups (Safirova and Walls 2004). This is also identified in the European Union. Non-teleworker men and women have lower monthly income compared to occasional, mobile, or home-based teleworkers (Rodríguez-Modroño and López-Igual 2021). This may be based on productivity gains justifying higher revenues, although initially, the change from face-to-face work to teleworking generated a negative shock in productivity.

Bick et al. (2020) argue that isolation led to difficulties when requesting and obtaining specific support from colleagues, generating delays at work. For this reason, they recommend establishing a structure for decision making and effective communication between the collaborators of a company, as has been practiced by large Chinese companies that have used teleworking. However, a randomized controlled experiment on teleworking was carried out in China at a large travel company, randomly assigning workers to a control group (traditional work method) and a treatment group (teleworking). A 13% performance increase was found for those who worked from home—nine points stemming from working more minutes per shift due to fewer breaks and sick days, and four more points related to increased calls per minute—than control workers; this was attributed to the quieter work environment. In addition, the performance of workers who remained in the office was not affected by the fact that some of their colleagues telecommuted (Bloom et al. 2015).

**Social Security**. Green (2009) reviewed information from many countries with widely differing institutional contexts and at varying stages of development. He found that workers rank job security (i.e., social security and job stability) consistently as the most important item in their preferences with respect to different aspects of work in almost all countries for which data are available. In a job, social security is linked to contractual conditions; therefore, changes in social security coverage are not assumed under the same contractual modality.

Working outside the facilities of an employer or contractor suggests a possible increase in ergonomic health risks and psychosocial risks due to isolation (ILO 2021). In fact, in a subsequent technical note by the World Health Organization (WHO), it was concluded that traditional workplaces may not comply with occupational health and safety standards (ILO and WHO 2021). In fact, in a subsequent technical note by the World Health Organization (WHO), it is concluded that traditional workplaces may not comply with occupational health and safety standards (ILO and WHO 2021), although the general conclusion of the study is:

> When organized and carried out properly, telework can be beneficial for mental health and social well-being. It can improve work–life balance, reduce time spent on commuting to the workplace, and offer opportunities for flexible work arrangements, all of which may promote mental health and social wellbeing. (ILO and WHO 2021, p. 7)

However, in the literature, it is mentioned that there could be a deficit in social security coverage in teleworkers working as freelancers, as well as other homeworkers, because they are treated as independent contractors (ILO 2021).

**Job stability**. There is some research on job stability and telework, but studies have primarily focused on how job stability affects job satisfaction in teleworking conditions rather than its level relative to other types of work. For example, in a job satisfaction survey of teleworkers in Mexico, it was estimated that 73.6% of those surveyed were satisfied with their job stability in this work modality (Ordóñez 2018). Eurofound and ILO (2019) find that length of employment and job stability are positively correlated with discretion and autonomy in workers, making them more likely to telework (Eurofound and ILO 2019).

In general, the ILO suggests that if workers freely telework, the benefits outweigh the costs (ILO 2021). However, it must be considered that during the pandemic, a large portion of workers worked from home because they had no other option. In any case, keeping

this caveat in mind, the literature points out that teleworking alters the traditional way of working, including the organization of firms that must move towards decentralized models providing broad autonomy to their workers (Lamond et al. 1998; ILO 2020c). Changes are also fostered in the way cities develop. The increasing availability of TIC foments labor productivity in some occupations and changes the costs relative to urban transportation systems. Thus, it could promote decentralization of cities with a significant impact on transportation, telecommunications, labor, and land-use policies (Nilles 1975). Such a change has a direct relationship with the quality of employment, usually measured through indices that use the characteristics of the jobs. This change is reviewed as part of this research.

## 3. Materials and Methods

### 3.1. Materials: Data

In this research, Brazil, Colombia, and Argentina were selected as reference countries to study the relationship between teleworking and job quality in Latin America. These countries are heterogeneous among themselves and have the first, third, and fourth largest population sizes in Latin America, respectively. In addition, these countries have labor surveys continuously collected which are available to the public[7].

This study uses databases of official labor market surveys of the statistical institutes of Colombia (Great Integrated Household Survey—GEIH, its acronym in Spanish), Argentina (Permanent Household Survey—EPH, its acronym in in Spanish), and Brazil (National Household Sample Survey—PNAD, its acronym in Portuguese). The point of interest is the employed population in urban areas[8] in annualized data for 2021[9]. It is important to note that for purposes of comparison between countries in this research, the common variables used in the analysis of the 3 countries were harmonized[10].

### 3.2. Job Quality Index Methodology

To study job quality, as mentioned above, multidimensional indices were constructed and analyzed. This can be achieved in different ways. The indices studied in the literature usually give equal weight to each component or dimension included, although the assignment of the weights of each characteristic generates controversy (Muñoz et al. 2011). To overcome this situation, in this research the job quality index was built using factorial analysis (a multivariate analysis technique) to study the structure of interdependencies between variables. This approach is especially useful for investigating complex or multidimensional relationships of a phenomenon (Aranzazu et al. 2007).

Factorial analysis includes several techniques. In this study, the factorial analysis of the total variance was applied wherein common factors were extracted, thus explaining the total variance. Specifically, we used a categorical component analysis, wherein quantitative and qualitative variables can be combined (López-Roldán and Fachelli 2015). This approach seeks factors as expressions of combinations of the original variables. It allows for the expression of the main differentiating factors and orders them hierarchically, thereby reducing the loss of information and gaining significance. Therefore, the combination of the components or dimensions of the index is created directly from the data. The weightings do not correspond to hypotheses or reflections of the researchers. Instead, they are derived from the dynamics of the data.

As previously mentioned, job quality indices in Latin America have previously focused on the components of income, social security coverage, job stability, and workday due to their conceptual importance and the availability of information in this regard. Therefore, this research utilizes these components, applying the factorial analysis technique for the construction of the index for national totals and for the groups of teleworkers and non-teleworkers within each country analyzed.

### 3.3. Impact Evaluation Methodology

To compare the conditions of the job quality index and each component among the interest groups, the method of impact evaluation is applied, specifically the matching technique. According to Gertler et al. (2016) "Matching essentially uses statistical techniques to construct an artificial comparison group. For every possible unit under treatment, it attempts to find a nontreatment unit (or set of nontreatment units) that has the most similar characteristics possible." (Gertler et al. 2016, p. 143).

For the purposes of this research, teleworkers are taken as the treatment group because they are workers who can work in the traditional way but are subjected to a new way of performing their functions. It seem that they "receive a treatment." Teleworkers are compared with a control group comprising other workers. The control group is constructed or selected from the general workers; each individual in the group of teleworkers is compared with an individual in the control group that is most similar to him or her, based on the available characteristics in the datasets.

The strategy used for matching is propensity score matching (PSM). PSM produces estimates for the difference between groups in each job quality component and the index for all treated observations that are matched with control observations. The PSM estimates the following:

$$\tau^{ATT} = E(Y_i(1)|D_i = 1) - E(Y_i(0)|D_i = 1) \tag{1}$$

where

$\tau^{ATT}$: average differences of the treatment on the treated (ATT)
$E(.)$: average or mathematical expectation
$Y_i$: potential outcomes (level of each job quality component)
$D_i$: group, denoted 1 for treatment and 2 for control.

As mentioned, to achieve this estimation, it is necessary to previously match the individuals of the treatment group with their peers in the control group considering their observable characteristics. For this purpose, this research uses the determinants of teleworking proposed by Oviedo-Gil and Cala (2022), that is, the number of children in the home, status of the spouse, educational level, work experience, and the size of the firm and the economic sector[11].

## 4. Results

The universe of interest corresponds to the working population in urban areas, in particular, the comparison between teleworkers[12] and other workers. The empirical approach begins with a descriptive analysis of the main components of job quality available for the countries under study, namely, labor income, social security, length of employment, and working hours[13]. Next, the estimates of the job quality index are presented. Finally, the comparisons calculated through the impact analysis are shown and analyzed, and what was obtained is contrasted with other studies.

Before considering the results, it is important to mention that this research faces some limitations. First, the variables used in the analysis correspond to observable characteristics of workers and jobs. Household surveys in Latin America do not capture subjective aspects such as job satisfaction in a standardized manner. Second, the job quality dimensions studied are those simultaneously available in Brazil, Argentina, and Colombia, and are susceptible to harmonization. Third, this study uses the teleworking approximation proposed by Oviedo-Gil and Cala (2022), which is robust, although it would be ideal to have a direct measurement as is found in studies conducted in Europe or the USA.

### 4.1. Average Comparison (Descriptive Analysis)

In a preliminary inspection of the data at a descriptive level, it is found that there are differences in each component of job quality observed between teleworkers and other workers. The heterogeneity in the characteristics of the workers can be explained by TLMS, which proposes restrictions on workers in moving between segments. A simple

comparison of means is presented in Table 2. In Colombia, teleworkers show lower levels of social security coverage (−18.5%) and have a shorter working day (−5.3%). In addition, they show higher labor income (23.9%) and greater job stability since they register higher lengths of employment (5.4%). The Argentine data suggest that teleworkers have greater social security coverage (19.1%), income (32.9%), and length of employment (4.9%), while they have shorter working hours (−14.7%) than non-teleworkers. In Brazil, the simple comparison of means indicates that teleworkers have a strong gap in social security coverage (−45.7%), shorter working hours (−14.8%), slightly shorter length of employment (−0.3%), and higher labor income (23.5%).

**Table 2.** Average values of job quality components for Argentina, Brazil, and Colombia, 2021.

| | Colombia | | | Argentina | | | Brazil | | |
|---|---|---|---|---|---|---|---|---|---|
| | **Teleworkers** | **Other Workers** | **Difference (%)** | **Teleworkers** | **Other Workers** | **Difference (%)** | **Teleworkers** | **Other Workers** | **Difference (%)** |
| Social security | 37.4% | 44.4% | −18.5% | 59.1% | 47.8% | 19.1% | 46.7% | 68.1% | −45.7% |
| Labor income | 1,407,567 | 1,071,278 | 23.9% | 63,790 | 42,813 | 32.9% | 3226 | 2468 | 23.5% |
| Length of employment | 3.0 | 2.9 | 5.4% | 3.5 | 3.3 | 4.9% | 2.7 | 2.7 | −0.3% |
| Workday | 43.4 | 45.6 | −5.3% | 30.4 | 34.9 | −14.7% | 34.6 | 39.7 | −14.8% |

Source: Own elaboration based on GEIH-DANE, EPH-Argentina, and PNAD-IBGE.

From this initial review, some trends are identified; specifically, there are gaps in labor formality measured as social security coverage for teleworkers in Brazil and Colombia. Furthermore, in all three countries, teleworkers report higher incomes and shorter working hours than other workers.

*4.2. Job Quality Index Results*

The main statistical moments of the distribution of job quality indices for Colombia, Argentina, and Brazil are graphed in Figure 1. The index was estimated separately for each country and the results are plotted for teleworkers (continuous line) and other workers (dashed line). Given that the variables used were harmonized, the graphical comparison between countries indicates that, in general, the job quality is higher in Argentina and similar between Colombia and Brazil; although, in Colombia the index starts at lower values.

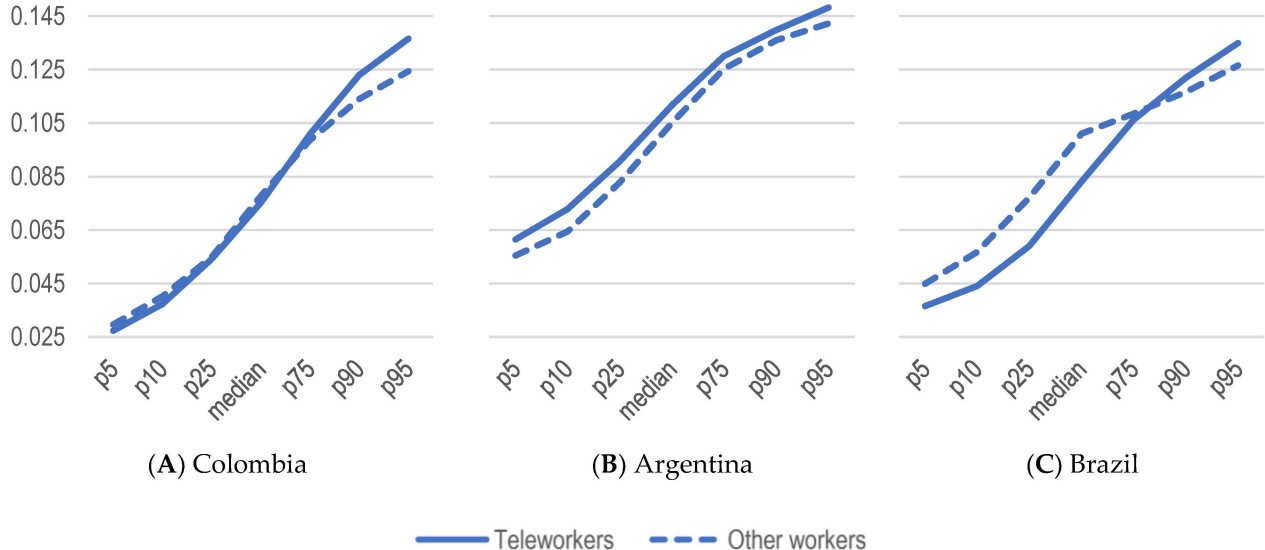

**(A)** Colombia **(B)** Argentina **(C)** Brazil

**Figure 1.** Main statistical moments of the distribution of the job quality indices estimation for Colombia, Argentina, and Brazil 2021. Source: Own elaboration based on GEIH-DANE, EPH-Argentina, and PNAD-IBGE.

Regarding the results within each country, the following stand out: in Colombia (panel A), between the 5th and 50th percentile (median), the index for teleworkers is lower than for other workers. Meanwhile, for values from the 75th percentile onwards, the behavior is reversed. However, the differences are small. The results for Argentina show that the index values are slightly higher for teleworkers throughout the distribution (panel B). Finally, in Brazil, the index values for teleworkers are visibly lower than for other workers in most of the distributions. Only from the 90th percentile onwards do teleworkers show a higher job quality index than other workers (panel C).

*4.3. Impact Evaluation Results*

The differences observed in the job quality index and its components between teleworkers and other workers in previous sections may not be statistically or analytically significant. For this reason, in order to strengthen the analysis of the differences, the PSM technique was used. This technique allows a comparison of teleworkers solely against other workers with similar characteristics, and not against other workers in general. It is relevant to point out that for each teleworker, it was possible to identify similar workers regarding observable characteristics to make the comparison. In other words, there was full common support. The results are summarized in Table 3.

4.3.1. Labor Income

Estimates show that in a controlled analysis, labor income is higher for teleworkers compared to other workers in Colombia and Argentina at 21.8% and 13.5%, respectively. In Brazil there are no significant differences. Here, the greater attention of firms toward work manifested as product or goals in teleworking could be related to an improvement in labor productivity that allow for better remunerations (Bloom et al. 2015; ILO 2020c). This is consistent with findings in California (USA) (Safirova and Walls 2004) and in the European Union.

Therefore, it seems necessary to diagnose whether or not workers could work toward goals if they have a teleworker profile that would make them much more likely to be successful in this modality, as some authors have suggested (Jaramillo and Bustamante 2011; Amigoni and Gurvis 2009).

4.3.2. Social Security Coverage

When reviewing the social security component, it is detected that teleworkers have less coverage: −15.9 percentage points (p.p.) in Brazil and −3.2 p.p. in Colombia. In Argentina, the difference is also negative, but not statistically significant. This was also found in the literature. There is a deficit in social security coverage for teleworkers, especially those working as freelancers. More specifically, according to ILO, in low- and middle-income countries almost all home-based workers (including teleworkers) do not have social security coverage for their jobs. The hypothesis that would explain this situation is that they are treated as independent contractors and their access to social security tends to be limited (ILO 2021).

This is a sensitive matter, as this characteristic of a position is highly valued by workers around the world (Green 2009). Being covered for risks associated with work allows workers to focus more on being productive and acquiring good work practices. Herein arises the need to adapt the structure of social security systems to facilitate access for this new type of worker, especially when they are self-employed.

**Table 3.** Differences between teleworkers and other workers in the job quality index and its components by PSM.

| | | Labor Income (Local Currency) | | Covered by Social Security (%) | | Length of Employment (Duration Range) | | Workday (Hours per Week) | | Job Quality Index | |
|---|---|---|---|---|---|---|---|---|---|---|---|
| Colombia[14] | Treated | 1,272,710 | | 33.2% | | 3.07 | | 42.8 | | 0.077 | |
| | Controls | 1,044,670 | | 36.4% | | 2.99 | | 43.8 | | 0.075 | |
| | Difference | 228,040 | ** | −3.2 | ** | 0.08 | ** | −1.0 | ** | 0.001 | * |
| | (%) diff | 21.8% | | −9 | | 2.5% | | −2.2% | | 1.6% | |
| Argentina[15] | Treated | 51,021 | | 52.5% | | 3.51 | | 27.0 | | 0.103 | |
| | Controls | 44,951 | | 53.7% | | 3.51 | | 32.0 | | 0.106 | |
| | Difference | 6070 | ** | −1.11 | | 0.00 | | −5.0 | ** | −0.003 | ** |
| | (%) diff | 13.5% | | −2 | | 0.0% | | −15.6% | | −2.5% | |
| Brazil[16] | Treated | 2429 | | 38.3% | | 2.74 | | 33.3 | | 0.079 | |
| | Controls | 2487 | | 54.2% | | 2.77 | | 39.2 | | 0.089 | |
| | Difference | −58 | | −15.9 | ** | −0.03 | ** | −5.9 | ** | −0.011 | ** |
| | (%) diff | −2.3% | | −29 | | −1.2% | | −15.0% | | −11.9% | |

Source: Own elaboration based on GEIH-DANE, EPH-INDEC, and PNAD-IBGE. Notes: ** statistically significant at 95%. * statistically significant at 90%.

### 4.3.3. Job Stability

In the component of length of employment that was used as proxy of job stability, teleworker status has a slight negative impact in Brazil (−1.2%), a positive impact in Colombia (2.5%), and no impact in Argentina. Job stability is also a valuable condition for workers. Nevertheless, there are only small differences between teleworkers and other comparable workers. This could be because teleworking has only recently been implemented in Latin America, so current teleworkers are traditional workers who switched to telework due to the requirements of the pandemic. Thus, they stay in the same jobs.

### 4.3.4. Workday

Finally, it is possible to determine that teleworkers have shorter working hours. In Brazil and Colombia, the average hours worked per week is close to official numerations[17], so shorter working hours is an aspect that could be read as a positive situation. The difference is larger in Brazil, with −5.9 less hours per week, while in Colombia the difference is only one hour per week.

Teleworkers in Argentina work fewer hours per week; however, their average hours worked per week is 27 h in contrast with 32 h for other workers. Both averages represent a third of the official amount. It is possible that a relevant portion of workers wanted to work more hours per week or that the main job is accompanied by a second job to complement labor availability. The INDEC in Argentina considers as underemployed those people who (i) work less than 35 h per week and (ii) are willing to work more hours. In 2021, according to official statistics, 55.7% of workers were full-time employed while 13.3% were underemployed. The latter group could put downward pressure on the statistics of hours worked per week.

Other studies related to the workday have found that teleworkers needed to sacrifice personal time to meet the requirements of their jobs during pandemic isolation. Nonetheless, workers appreciate flexibility in the working day. This, plus possibilities of increasing productivity, could conduce workers and firms to optimize work hours to take advantage of these new opportunities.

### 4.3.5. Job Quality Index

PSM was also applied to the job quality index, which summarizes the behavior of its components. The results obtained indicate that teleworkers show lower average index values in Brazil, equivalent to −11.9% compared with other workers. This means there is a relevant deficit in job quality for teleworkers in this country. For Argentina and Colombia there is also a statistically significant difference, but its magnitude is too small to be analytically important at −2.5% and 1.6%, respectively.

## 5. Discussion and Conclusions

Technological progress and global isolation measures in 2020 have led society to implement a new way of working, namely, teleworking. Furthermore, it is predicted that teleworking is here to stay as it is a viable alternative for certain firms and occupations (Guyot and Sawhill 2020). Thus, there has been a transformation in the world of work that needs to be measured, modeled, and monitored, since it could directly affect a variety of socioeconomic issues. A research gap is detected regarding the incidence of teleworking in job quality, labor participation, economic growth, productivity, and quality of life, among other factors. To contribute to closing this gap, this research studies what is meant by teleworking, what is meant by job quality, and how teleworking is related to job quality. This facilitates the identification of lines of public policy to be recommended to take advantage of the potential advantages of teleworking and address its possible disadvantages.

The conceptual review indicates that teleworking is a remote work or "home work" strategy that relies on ICT for its operation. In addition, it is found that job quality is a multidimensional concept which includes several job characteristics that could affect the well-being of workers. Although there is no agreement on which characteristics to consider,

in LAC, working hours, social security coverage, labor income, and job stability have been frequently studied. This could be summarized in a job quality index.

This study builds a job quality index that summarizes the dimensions of income, social security coverage, working hours, and job stability measured as length of employment. The index was built using factorial analysis; instead of assigning a priori weights to each component, the weights arise from the data themselves, thus excluding possible subjectivities of the authors and considering the characteristics of each country. Given that the variables used were harmonized, the simple comparison between countries indicates that the job quality is higher in Argentina and similar between Colombia and Brazil, although in Colombia, the index starts at lower values. Regarding the results within each country, it is found that the group of teleworkers and the group of other workers show small differences in a descriptive contrast.

The construction and measurement of the job quality index in this study could be replicated for other Latin American countries since the information incorporated is commonly collected in the countries of the region. However, an index reduces or hides the variability of its components because it balances positive and negative aspects of a job. Therefore, studying each dimension separately remains relevant.

This research analyzes each job quality component and an associated comparison-built index from an impact approach, which allows for the comparison of teleworkers solely with other workers similar to them in regard to their observable characteristics. This is achieved using the PSM impact evaluation technique. Estimates indicate the following regarding teleworkers:

(i)    They receive higher income in Colombia (21.8%) and Argentina (13.5%).
(ii)   They have less social security coverage in Brazil (−15.9 p.p.) and Colombia (−3.2 p.p.).
(iii)  They have approximately the same job stability.
(iv)   They work less hours per week in Brazil (−5.9 h), Argentina (−5.0 h), and Colombia (−1.0 h), which is positive for Brazil and Colombia, where the average working day is close to the official one, but may be negative for Argentina, where there is a deficit of hours worked.
(v)    They exhibit a job quality index with lower average values in Brazil (−11.9%), while Colombia and Argentina present analytically equal values to other comparable workers.

In Brazil and Colombia, there are significant gaps in social security coverage for teleworkers. Closing these gaps should be a public policy objective, given the importance of insurance for individual and social well-being. An update in the methods of access to social security is required, recognizing the characteristics of teleworkers and their way of working, which can be described as self-employed. It would also be useful to implement support for linking and paying to the system, considering that in countries such as Colombia, teleworkers show higher relative income compared to other comparable workers.

The differences found in working hours per week in Colombia and Brazil suggest that some type of time optimization is taking place. The question arises whether this is genuinely happening and how. Perhaps research case studies with an in-depth analysis could provide good practices, which could be understood and disseminated in favor of improvements in the labor productivity of those who use teleworking, both firms and workers.

There is heterogeneity in jobs in the LAC countries analyzed, as indicated by objective comparisons made in this study. This can be explained by the theory of labor market segmentation (TLMS) which suggests that the labor market is divided into a formal segment with high productivity and income and an informal one with precarious working and productive conditions. There is evidence in the literature that this segmentation occurs in Argentina, Colombia, and Brazil.

Several authors point out that teleworking implies changes in the organizational and administrative structure of firms that lead to a decentralized operation with great autonomy for teleworkers. In addition, the implementation of successful teleworking requires adequate profiles of the personnel; for example, the ability to use ICT and soft skills such as self-regulation and discipline to work toward goals or results are needed.

All this would indicate that teleworking is mostly associated with the formal sector of the economy and, therefore, good working conditions associated with it would be expected. However, empirical studies and results have shown mixed results.

The results show that teleworking is associated with favorable conditions in some dimensions of job quality, so it could be promoted by the countries of reference. For this, it must be taken into account that firms require support to move towards decentralized administration models where there is less surveillance of workers, leading to flexible hours and work for goals or results. In addition, training in ICT and the soft skills necessary to work toward results with autonomy must be provided. Thus, public policy for teleworking must consider these needs identified in both the supply and the demand for labor.

To understand the quality of work in Latin America in a more comprehensive way, it is recommended that statistical institutes in the region include subjective characteristics associated with well-being and job satisfaction in their surveys. Colombia includes a set of questions in this regard, but the absence of information in other countries inhibits the comparison. In fact, it would be valuable to have standardized guidelines from entities such as the ILO so that measurement and comparison are adequate.

**Author Contributions:** Conceptualization, Y.M.O.-G.; Formal analysis, Y.M.O.-G.; Investigation, Y.M.O.-G.; Methodology, Y.M.O.-G.; Supervision, F.E.C.V.; Writing—original draft, Y.M.O.-G.; Writing—review & editing, Y.M.O.-G. and F.E.C.V. All authors have read and agreed to the published version of the manuscript.

**Funding:** This research received no external funding.

**Institutional Review Board Statement:** Not applicable.

**Informed Consent Statement:** Not applicable.

**Data Availability Statement:** Data in this study can be obtained from https://www.indec.gob.ar/indec/web/Institucional-Indec-BasesDeDatos-1 (accessed on 20 August 2022) (Argentina); https://www.ibge.gov.br/estatisticas/sociais/populacao/9173-pesquisa-nacional-por-amostra-de-domicilios-continua-trimestral.html?t=microdados (accessed on 20 August 2022) (Brazil); https://microdatos.dane.gov.co/index.php/catalog/701/get-microdata (accessed on 20 August 2022) (Colombia).

**Conflicts of Interest:** The authors declare no conflict of interest.

## Notes

[1] Some interesting studies in this regard are Azarbouyeh and Naini (2014) and Rodríguez-Modroño and López-Igual (2021). Azarbouyeh and Naini (2014) found that teleworking has a significant positive relationship with the components of quality of work life; this affected all types of workers in a similar way. However, Rodríguez-Modroño and López-Igual (2021) found that job quality among teleworkers varies according to the intensity of ICT use, gender, and type of employment.

[2] Own translation.

[3] https://www.argentina.gob.ar/trabajo/teletrabajo-0/teletrabajo-y-contrato-de-teletrabajo—Own translation, accessed on 10 March 2023. Teleworking in Argentina is regulated by (Law 27555 2020) that stipulates the contractual conditions that must be met, working hours, rights and duties, training, benefits, among others.

[4] Own translation. The regulation of teleworking in Brazil is found mainly in the (Law 13467 2017).

[5] The dimensions are safety and ethics of employment, income, and benefits from employment, working time and work-life balance, security of employment and social protection, social dialogue, skills development and training, and employment-related relationships and work motivation.

[6] The elements are employment opportunities, adequate earnings and productive work, decent working time, combining work, family, and personal life, work that should be abolished, stability and security of work, equal opportunity and treatment in employment, safe work environment, social security, and social dialogue, employers', and workers' representation. Consequently, this approach to job quality is focused on monitoring aggregated decent work at the country level.

[7] The databases are openly available at: https://www.indec.gob.ar/indec/web/Institucional-Indec-BasesDeDatos-1 (Argentina); https://www.ibge.gov.br/estatisticas/sociais/populacao/9173-pesquisa-nacional-por-amostra-de-domicilios-continua-trimestral.html?t=microdados (Brazil); https://microdatos.dane.gov.co/index.php/catalog/701/get-microdata (Colombia).

[8] This was to achieve comparability with Argentina, whose survey is limited to urban areas.

9.  For Brazil, the first three quarters of the year are considered, since an error was detected in education variables in the fourth quarter that was reported to the Brazilian Statistics Institute. However, the available databases are robust enough to run the analysis on them.
10.  The software used for the calculations was Stata 17 SE.
11.  This is entered in a logit model that feeds the matching method used—in this case, "nearest neighbor", which consists of selecting for each individual in the "treatment" group five "control" individuals with the closest propensity score.
12.  According to the literature review, this study takes the definition of teleworkers as those who practice "*the use of ICT—such as smartphones, tablets, laptops and desktop computers—for the purposes of work outside the employer's premises*" (MarcadorDePosición3) Page 5, and it is measured according to (Oviedo-Gil and Cala 2022). According to estimates, in 2021, 6.6% of all urban workers were teleworkers in Colombia, 8.4% in Argentina, and 5.5% in Brazil.
13.  The variables are defined as follows: (i) labor income corresponds to the aggregation of income created by each national statistical institute; (ii) social security is a variable that indicates whether or not a worker is contributing to the pension system; (iii) length of employment indicates how long a worker has been in their current position (less than 1 month, 1 month to 1 year, more than 1 year to 5 years, or 5 years or more); and (iv) workday is the number of hours worked per week.
14.  Number of observations 197,926; LR chi2(11) 12091.22; Prob > chi2 0.000; Pseudo R2 0.1114.
15.  Number of observations 62,569; LR chi2(11) 3231.99; Prob > chi2 0.000; Pseudo R2 0.1328.
16.  Number of observations 266,568; LR chi2(11) 18725.70; Prob > chi2 0.000; Pseudo R2 0.1505.
17.  The maximum working day allowed in Argentina and Colombia is 48 h per week, while in Brazil it is 44 h per week.

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
