# Peer review of "Teleworking and Job Quality in Latin American Countries: A Comparison from an Impact Approach in 2021"

_socsci, doi:10.3390/socsci12040253_

Round 1
Reviewer 1 Report
Recommendation: Accept with Minor Revision
Dear author/s,
Thank you for the opportunity to review this manuscript. I hope you find my comments useful as you consider revising the paper. The topic is fitting with the aim and scope of the Journal. I hope this review provides some useful feedback and wish you the best of luck with the development of this paper!
Additional Questions:
1. Originality: Does the paper contain new and significant information adequate to justify publication?: Yes. he authors identified an interesting topic and provided significant information on the relation of “Teleworking and Job Quality in Latin American Countries: A Comparison from an Impact Approach in 2021”. However, there are some issues that need clarification. More importantly, the discussion regarding the research gaps is missing!
2. Relationship to Literature: Does the paper demonstrate an adequate understanding of the relevant literature in the field and cite an appropriate range of literature sources? Is any significant work ignored?: The literature review section is fine.
3. Methodology: Is the paper's argument built on an appropriate base of theory, concepts, or other ideas? Has the research or equivalent intellectual work on which the paper is based been well designed? Are the methods employed appropriate?: The paper is well structured and follows the standards.
4. Results: Are results presented clearly and analysed appropriately? Do the conclusions adequately tie together the other elements of the paper?: The results section is fine.
5. Implications for research, practice and/or society: Does the paper identify clearly any implications for research, practice and/or society? Does the paper bridge the gap between theory and practice? How can the research be used in practice (economic and commercial impact), in teaching, to influence public policy, in research (contributing to the body of knowledge)? What is the impact upon society (influencing public attitudes, affecting quality of life)? Are these implications consistent with the findings and conclusions of the paper?: The implications part missing.
6. Quality of Communication: Does the paper clearly express its case, measured against the technical language of the field and the expected knowledge of the journal's readership? Has attention been paid to the clarity of expression and readability, such as sentence structure, jargon use, acronyms, etc.: A professional review of the language is strongly suggested because several parts of the text are unclear.
Reviewer 2 Report
The paper addresses a current topic and is generally quite well structured, but I consider it needs some improvement.
Introduction: I consider that the research gap and research questions should be outlined.
Teleworking and its Relationship with Job Quality:
Although the topic is a new one, very recent bibliographic sources (after 2020) are few. I consider that new bibliographic resources should be identified and added.
In-text citations should be revised:
- as national definition for teleworking in Latin American countries (see Error! Reference source not found.) - Lines 85-86
- Steffgen, Sischka, & Fernandez, 2020 (Lines 92-93) - Steffgen and al., 2020
- same problems: Lines 130-131; Lines 142-143; Line 156; Lines 179-180; Line 186; Line 256.
Materials and Methods:
The research methods used should not be presented at a basic level (for example, factorial analysis).
Discussion and Conclusions:
The authors should compare their findings with the results of similar studies.
In this section the limits of the research should be highlighted.
References:
References should be formatted according to the journal's accepted style.
